# Correlated signatures of social behavior in cerebellum and anterior cingulate cortex

Sung Won Hur[1,2†], Karen Safaryan[1†], Long Yang[3], Hugh T Blair[4], Sotiris C Masmanidis[3], Paul J Mathews[2,5], Daniel Aharoni[1*], Peyman Golshani[1*]

[1]Department of Neurology, DGSOM, University of California Los Angeles, Los Angeles, United States; [2]The Lundquist Institute for Biomedical Innovation, Harbor-UCLA Medical Center, Torrance, United States; [3]Department of Neurobiology, University of California Los Angeles, Los Angeles, United States; [4]Department of Psychology, University of California Los Angeles, Los Angeles, United States; [5]Department of Neurology, Harbor-UCLA Medical Center, Torrance, United States

*For correspondence:
daharoni@mednet.ucla.edu (DA);
PGolshani@mednet.ucla.edu
(PG)

†These authors contributed
equally to this work

Competing interest: The authors
declare that no competing
interests exist.

Reviewing Editor: Brice
Bathellier, CNRS, France

**Abstract** The cerebellum has been implicated in the regulation of social behavior. Its influence is thought to arise from communication, via the thalamus, to forebrain regions integral in the expression of social interactions, including the anterior cingulate cortex (ACC). However, the signals encoded or the nature of the communication between the cerebellum and these brain regions is poorly understood. Here, we describe an approach that overcomes technical challenges in exploring the coordination of distant brain regions at high temporal and spatial resolution during social behavior. We developed the E-Scope, an electrophysiology-integrated miniature microscope, to synchronously measure extracellular electrical activity in the cerebellum along with calcium imaging of the ACC. This single coaxial cable device combined these data streams to provide a powerful tool to monitor the activity of distant brain regions in freely behaving animals. During social behavior, we recorded the spike timing of multiple single units in cerebellar right Crus I (RCrus I) Purkinje cells (PCs) or dentate nucleus (DN) neurons while synchronously imaging calcium transients in contralateral ACC neurons. We found that during social interactions a significant subpopulation of cerebellar PCs were robustly inhibited, while most modulated neurons in the DN were activated, and their activity was correlated with positively modulated ACC neurons. These distinctions largely disappeared when only non-social epochs were analyzed suggesting that cerebellar-cortical interactions were behaviorally specific. Our work provides new insights into the complexity of cerebellar activation and co-modulation of the ACC during social behavior and a valuable open-source tool for simultaneous, multimodal recordings in freely behaving mice.

## eLife assessment

Based on a technological advance which couples onboard calcium imaging with in vivo electrophysiology in freely behaving mice, this **important** work presents data about the modulation of some long range brain activity correlations during social interactions. **Solid** evidence shows that neural activity across cerebellum and cingulate cortex is more correlated during social behaviors than during non-social epochs. This study is of interest for a broad range of neurophysiologists.

**eLife digest** Social behaviour is important for many animals, especially humans. It governs interactions between individuals and groups. One of the regions involved in social behaviour is the cerebellum, a part of the brain commonly known for controlling movement. It is likely that the cerebellum connects and influences other socially important areas in the brain, such as the anterior cingulate cortex. How exactly these regions communicate during social interaction is not well understood.

One of the challenges studying communication between areas in the brain has been a lack of tools that can measure neural activity in multiple regions at once. To address this problem, Hur et al. developed a device called the E-Scope. The E-Scope can measure brain activity from two places in the brain at the same time. It can simultaneously record imaging and electrophysiological data of the different neurons. It is also small enough to be attached to animals without inhibiting their movements.

Hur et al. tested the E-Scope by studying neurons in two regions of the cerebellum, called the right Crus I and the dentate nucleus, and in the anterior cingulate cortex during social interactions in mice. The E-Scope recorded from the animals as they interacted with other mice and compared them with those in mice that interacted with objects.

During social interactions, Purkinje cells in the right Crus I were mostly less active, while neurons in the dentate nucleus and anterior cingulate cortex became overall more active. These results suggest that communication between the cerebellum and the anterior cingulate cortex is an important part of how the mouse brain coordinates social behaviour.

The study of Hur et al. deepens our understanding of the function of the cerebellum in social behaviour. The E-Scope is an openly available tool to allow researchers to record communication between remote brain areas in small animals. This could be important to researchers trying to understand conditions like autism, which can involve difficulties in social interaction, or injuries to the cerebellum resulting in personality changes.

## Introduction

The cerebellum, a brain region traditionally associated with motor coordination, has been increasingly recognized as a region that participates in the coordination of social cognition and behavior. Early studies showed that cerebellar stroke or injury could result in personality changes, blunted affect, and even antisocial behavior (*Schmahmann and Sherman, 1998*). More recent studies using neuroimaging and behavioral assays indicate the cerebellum contributes to elements of social cognition including 'mentalizing' or 'mirroring' (*Van Overwalle et al., 2020*).

Studies in animal models have strengthened the link between the cerebellum and social behavior. Modulating the neural activity of cerebellar PCs through the insertion of disease related gene mutations or chemogenetic alterations of firing rate reduced social interactions (*Badura et al., 2018*; *Kelly et al., 2020*). Moreover, optogenetic suppression of cerebellar output to the forebrain or midbrain, especially cerebellar neurons projecting to the ventral tegmental area (VTA), decreased social interest (*Carta et al., 2019*). Among all cerebellar regions, modulation of the RCrus I of the cerebellar cortex had the most significant impact on the expression of social behavior (*Stoodley et al., 2017*).

However, it has yet to be established how the cerebellar output of the RCrus I region contributes to social behavior. It is likely that the cerebellum, through cerebellar communication with forebrain areas like the ACC, provides signals important in coordinating aspects of social interaction. These socially relevant brain regions have known multisynaptic connectivity, in which the RCrus I PC axons project to the DN, which in turn innervates the ventromedial thalamic neurons (*Badura et al., 2018*; *Kelly et al., 2020*). The cerebellum's interactions with the ACC are of interest as targeted lesions to it in non-human primates and most recently, selective gene deletion in mice, decreased sociability and the recognition of social and emotional cues (*Devinsky et al., 1995*; *Guo et al., 2019*; *Hadland et al., 2003*). While both the cerebellum and ACC play important roles in the regulation of social behavior, how these regions interact at the neural and systems levels to drive social behavior is still poorly understood.

Here, we developed the E-Scope, a novel device that can perform simultaneous electrophysiological and calcium imaging recordings of cerebellum and ACC, respectively. We have, for the first time, recorded the activity of RCrus I PC, or DN neurons in synchrony with recordings from large

populations of ACC neurons during social interaction. Our results demonstrate that PCs and DN neurons are mostly antagonistically modulated by social interaction with most PCs inhibited and DN neurons excited during interaction epochs. Furthermore, we find that there is a higher correlation between cerebellar and ACC neurons that are similarly excited or inhibited by social interaction than those modulated in an opposing manner, but that these differences disappear during non-social bouts. This highlights brain-state-specific modulation of social circuits across distant brain regions, critical for social interaction.

## Results

### E-Scope: A miniaturized calcium imaging microscope with integrated dense electrode technology for the synchronous acquisition of neural activity across distant regions of the brain

Limitations in the spatial and temporal resolution of current neural monitoring technologies have hampered our ability to systematically examine how interconnected brain regions communicate to control complex behaviors. To address these limitations, we have developed a novel device-based off the open-sourced UCLA Miniscope to synchronously measure single-cell activity at or near spike-time resolution across distant brain regions in freely behaving mice. We have integrated a miniatured microscope, which performs calcium imaging, with dense electrode electrophysiological recording, allowing simultaneous recordings from two remote brain regions in a freely behaving mouse (*Figure 1A*). This 'E-Scope' uses a single coaxial cable that minimally impacts animal movement to integrate both imaging and electrophysiological data streams along with the Miniscope's power. The electrophysiological component of this dual neural activity monitoring system is especially advantageous for brain regions with high basal firing rate neurons, such as the cerebellum, where activity may be obscured or not amenable to calcium imaging.

The E-Scope builds off our previous open-source wired Version 3 miniaturized microscope (Miniscope V3). Using its modular capability, we incorporated 32 channel electrophysiology sampling capabilities with the ability to support various configurations and geometries of silicon probes and tetrodes. The entire assembly weighs less than 4.5 g. For performing electrophysiological recordings, we used a two-shank, 64 channel silicon probe (64 H with 32 channels activated) (*Yang et al., 2020*).

We used the E-Scope to investigate correlations in the activity of RCrus I cerebellar PCs or DN neurons with population activity of contralateral ACC neurons during social behavior. To monitor ACC population activity, we injected an AAV virus to express the genetically encoded calcium indicator GCaMP6f in the left ACC. After a one-week recovery period, a 1 mm diameter GRIN lens (Inscopix; PN 130–000143) was implanted at the site of injection at a 10° angle. After a subsequent 2 weeks, the V3 Miniscope baseplate was implanted over the ACC injection site. Animals were introduced to the 48 × 48 cm behavioral arena and gradually acclimated to the weight of the scope by using dummy scopes of increasing weight over a 7–10 day period. Silicon-based, dense-electrode probes (*Yang et al., 2020*) were implanted and affixed into the RCrus I PC layer or DN and secured into place with dental cement. We made the first set of recordings ~24 hr after implantation of the probes (*Figure 1B and C*). Prior to implantation, the probes were coated with DiI (#C7000, Thermo Fisher Scientific) to enable post hoc confirmation of the probe's anatomical location (*Figure 1C*). For each mouse, we recorded approximately 200 ACC neurons using calcium imaging (*Figure 1D*) and 1–9 cerebellar neurons using dense electrode arrays (*Figure 1E*). The freeware analysis program Kilosort2/Phy2 was used to isolate single PC and DN units from recordings (*Figure 1F and G*). P-sort open-source software was used for PCs, which were identified by the presence of complex spikes (CS) along with the simple spikes (SS) in the recordings (*Figure 1G*; *Sedaghat-Nejad et al., 2021*).

### E-Scope recordings of cerebellar activity during social and object interactions

Implanted subject animals were placed into the behavioral arena for 1 min before the introduction of either another novel male C57Bl/6 mouse or a 3D-printed object (cone or cube) for 7 min. Subsequently, the animal was removed from the behavior arena for 1 min while the arena was cleaned. The recorded animal was then reintroduced to the arena and introduced to a novel mouse or object for another 7 min in a counterbalanced manner. Proximity to each other's body was used it as a measure

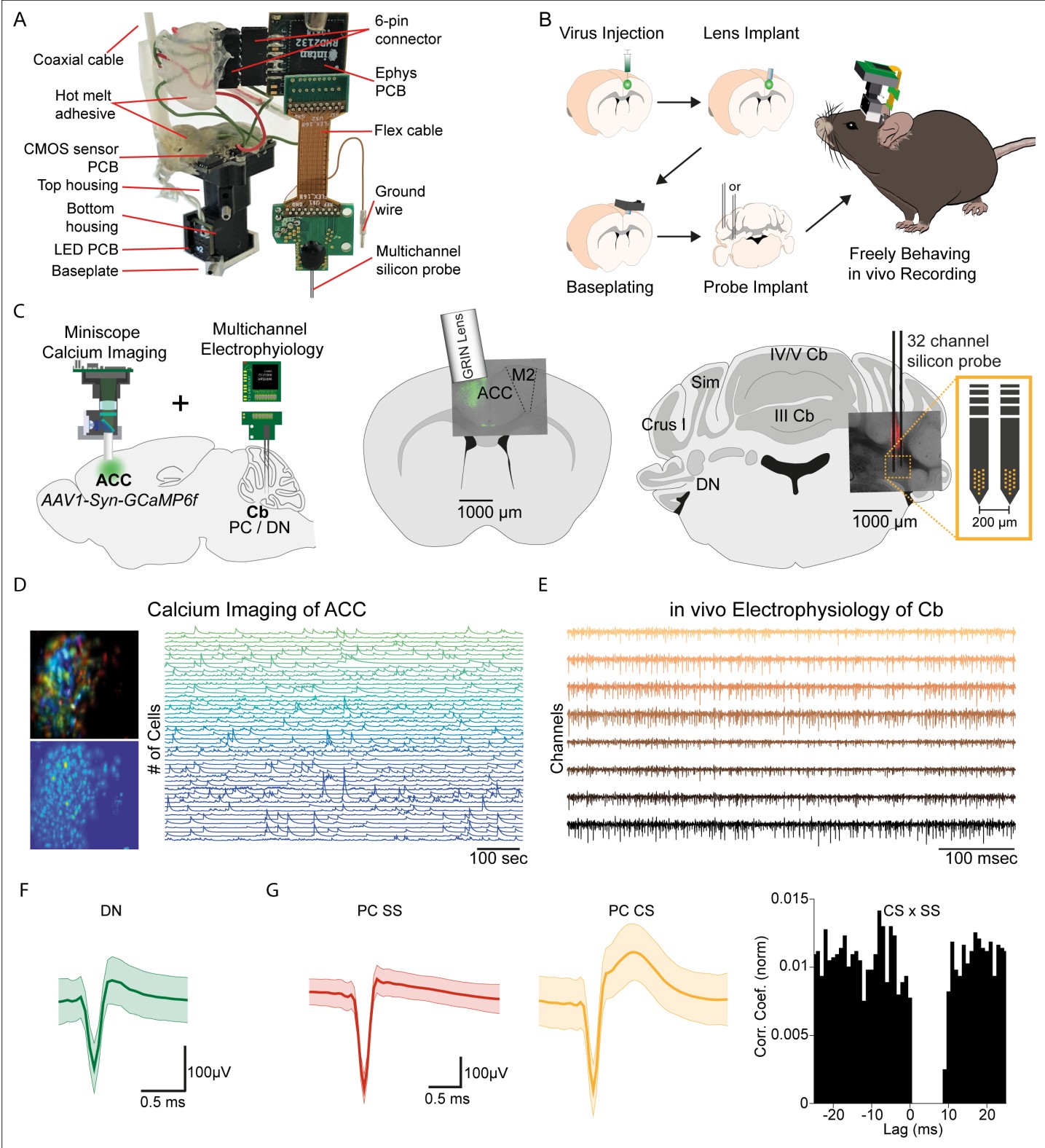

**Figure 1.** E-Scope: An integrated device allowing synchronous calcium imaging of anterior cingulate cortex and electrophysiological recordings in cerebellum. (**A**) Photograph of E-Scope hardware. The multichannel silicon probe (32 channels) connects to the custom Ephys PCB. The Ephys PCB is connected to the CMOS sensor printed circuit board (PCB) of the Miniscope via a 6-pin connector. The electrophysiology and image data streams are both conveyed through the coaxial cable. (**B**) Illustration of the process for implanting the E-Scope. (**C**) Illustrations and photomicrographs showing the location of AAV1-Syn-GCaMP6f virus injection in anterior cingulate cortex (ACC) (*left, mid*) and multichannel probe implant in the dentate nucleus of

**Figure 1 continued**

the cerebellum (*left, right*). (**D**) Pseudo-color (*top left*) and averaged activity heatmap from calcium imaging ACC neurons segmented using CNMF-E (*bottom left*). Calcium signals from neurons are shown on the left (*right*). (**E**) in vivo extracellular electrophysiology recording of Purkinje cells (PCs) in the cerebellum (Cb). (**F**) Average spike waveform of a dentate nucleus (DN) neuron. (**G**) Average simple spike (SS) waveform (*left*) and average complex spike (CS) waveform (*mid*) of a PC. Cross-correlogram of simple spikes and complex spikes shows the pause in simple spike activity after a complex spike (*right*).

The online version of this article includes the following figure supplement(s) for figure 1:

**Figure supplement 1.** E-Scope's hardware multiplexed data flow and Purkinje cell recording probe location.

of social interaction. Behaviors other than nose-to-nose, nose-to rear, or nose-to-body interactions were disregarded in the analysis. Recorded animals made more contact with the other mouse than with the object (***Figure 2A***), suggesting a normal preference for social contact with the E-Scope attached.

We recorded SS from a total of 28 PCs during both social and object interaction (***Figure 2B and G***). We used receiver operating characteristics (ROC) analysis to measure the area under the curve (AUC) value which quantified the overlap of activity with binarized vectors characterizing whether the animal was socially interacting or not interacting with the other animal; these values were in turn compared to circularly shuffled controls to identify units significantly modulated by social or object interaction. Nearly one-third of the PCs (28.5%, n=8) were robustly inhibited by social interaction (***Figure 2D, E and G***) while only three PCs (10.7%) were excited during social interaction (***Figure 2D***). The remaining non-significant (nsPC) units were not modulated (***Figure 2F***). Neurons inhibited by social interaction were inhibited for 1–3 s (***Figure 2C and E***), which was far longer than the SS pause is typically induced by CSs (up to ~40 ms) (***Figure 1G***). The same procedure revealed only 2 significantly modulated cells (7.1%) in both object interaction excited and inhibited groups (***Figure 2—figure supplement 1***).

Conversely, most DN neurons significantly modulated by social interaction were excited. Out of 99 DN neurons recorded, 20 neurons (20.2%) were excited during social interaction (Soc + DN), 12 neurons (12.1%) were inhibited (Soc– DN) and the remaining 67 not significantly modulated (nsDN) (***Figure 2I–N***). The excitation was more robust than inhibition and firing rates were increased for 2–4 s post interaction onset. The DN cells recorded in the object interaction sessions (n=85) showed smaller responses compared to the social interaction with 10 (11.7%) and 6 (7.05%) excited and inhibited units, respectively (***Figure 2—figure supplement 2***).

Cerebellar PCs have been reported to have different modules that operate at different baseline SS frequencies (***Zhou et al., 2014***). These modules have also been shown to modulate bidirectionally during learning (***De Zeeuw, 2021***). We, therefore, analyzed the mean firing rate during non-social events for PCs and DN neurons. DN neurons did not show any difference ($F$=1.589, p=0.2172) in mean firing rate between Soc+ and Soc– groups and the bimodal rate distribution of the DN neurons were not related to the waveform properties (***Figure 2—figure supplement 3A, B***). However, PCs showed a clear difference ($F$=5.742, p=0.04) in mean firing rate between the two socially responding types (***Figure 2—figure supplement 3C,D***).

Social and object interactions are often coupled with locomotion and head rotational changes, which are especially relevant to cerebellar activity. Thus, we further investigated the possibility that PCs and DN neurons encode movement-related activity. First, we examined the body speed and angular head velocity at the onset of social interaction. In the sessions recorded, we found mice tended to reduce their speed and head angular velocity upon interaction (***Figure 2—figure supplement 4A, B***). Analysis of the Soc+ and Soc– PCs as well as DN neurons showed no significant change in activity upon onset (***Figure 2—figure supplement 4C, D***) or offset (***Figure 2—figure supplement 4E, F***) of locomotion. In addition, head rotation onset had no relationship to PC or DN activity in Soc+ or Soc– groups (***Figure 2—figure supplement 4G, H***). Therefore, the activity changes we observed in cerebellar neurons at the onset of social interactions are unlikely to be related to motor aspects of behavior during social interactions. Overall, the electrophysiological responses in the cerebellum during social interaction are predominantly composed of PC inhibition and DN excitation. These results are consistent with the cerebellar anatomical circuit, where simultaneously decreased PC activity would be predicted to disinhibit DNs, thus resulting in excitatory output from the cerebellum to downstream brain areas (***Lee et al., 2015***).

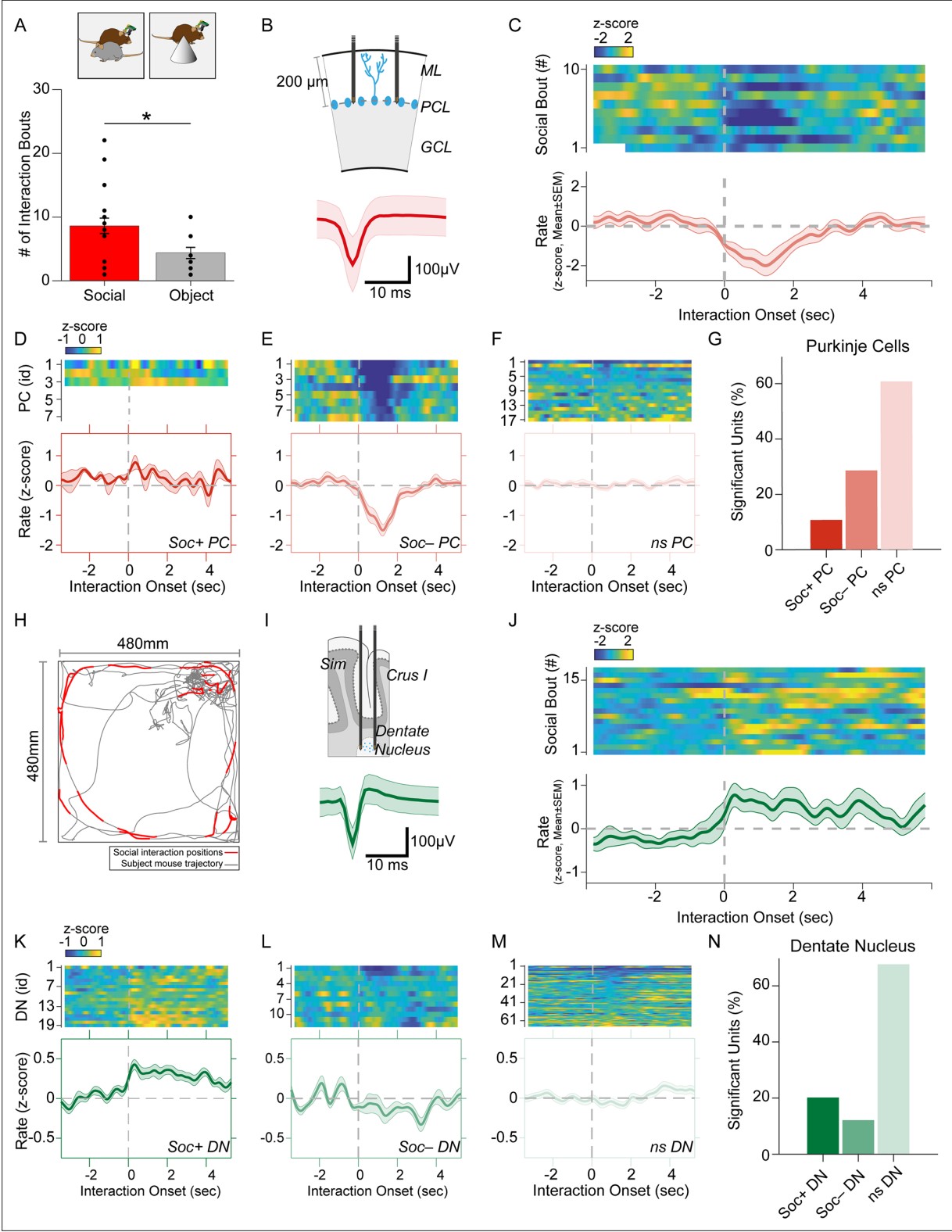

**Figure 2.** Purkinje cell and dentate nucleus neuron activity patterns during social behavior. (**A**) Graph of the number of interaction bouts between the recorded subject mouse and a novel target mouse or object (two-sided Wilcoxon rank sum test; p=0.0245, Z=2.2485) (**B**) Illustration of probe location in the Purkinje cells (PC) layer (*top*). Average simple spike waveform of a PC (*bottom*). (**C**) Heatmap of normalized (Z-score) firing rates of a Soc– PC neuron aligned to the onset of social interaction shown for 10 interaction epochs (*top*). The mean normalized firing rate across all interaction sessions shown above (*bottom*). (**D, E, F**) Average activity of three Soc+ PCs (**D**), 10 Soc– PCs (**E**), and 17 ns PCs (**F**). The mean activity of each group is shown

*Figure 2 continued on next page*

*Figure 2 continued*

below each heat map. (**G**) Proportion of Soc+, Soc–, and ns PCs in the recorded population. (**H**) Trajectory of subject mouse (gray) with social interaction locations indicated in red within the social interaction arena (480 × 480 mm). (**I**) Illustration of probe location for dentate nucleus (DN) recordings (*top*). Average spike waveform of a DN neuron (*bottom*). (**J**) Normalized firing rates of a Soc+ DN neuron aligned to the onset of social interaction shown for 18 interaction epochs (*top*). The mean normalized firing rate across all interaction sessions shown above. (**K, L, M**) Average activity of 19 Soc+ DN neurons (**K**), 10 Soc– DN neurons (**E**), and 63 ns DN neurons (**F**). The mean activity of each group is shown below each heat map. (**N**) Proportion of Soc+, Soc–, and ns DNs in the recorded population.

The online version of this article includes the following figure supplement(s) for figure 2:

**Figure supplement 1.** Object-evoked Purkinje cell responses differ from social-evoked responses.

**Figure supplement 2.** Object-evoked dentate nucleus neuron responses differ from social-evoked responses.

**Figure supplement 3.** Bimodality of socially excited and inhibited Purkinje cell simple spike but not dentate nucleus neurons.

**Figure supplement 4.** Socially excited and socially inhibited Purkinje cell and dentate nucleus activity are not related to locomotion speed.

## E-Scope calcium imaging of ACC neuron activity during social and object interaction

To determine how the cerebellar neurons communicate with ACC neurons during social interaction, we performed calcium imaging of ACC neurons (*Figure 3A*) synchronously with the electrophysiological recordings in the cerebellum discussed above using the E-Scope. We again used ROC analysis and circularly shuffled controls to determine whether ACC neurons were excited, inhibited, or not significantly modulated by social or object interactions (*Figure 3B–F*, *Figure 3—figure supplement 1*). We recorded 4868 and 3149 neurons during social and object interaction sessions, respectively. Of the recorded cells, 3.6% of ACC neurons were socially excited (Soc + ACC; *Figure 3B, D and F*), and 5.2% were socially inhibited (Soc– ACC; *Figure 3C, E and F*). These proportions were greater than what would be expected by chance given the criteria for AUC significance (p<0.025). Conversely, 4.1% and 8.4% of ACC neurons were excited or inhibited by object interactions, respectively (*Figure 3G and H*, *Figure 3—figure supplement 1*). The overlap of neurons significantly excited or inhibited by both social and object interactions was 0.1% and 0.4% of all neurons, respectively. Therefore, a significant proportion of non-overlapping ACC neurons were excited or inhibited by social and object interactions.

Finally, we assessed whether ACC neuron activity is correlated with locomotion or changes in head angular momentum. Similar to our observation in the cerebellum, ACC activity showed no significant correlation with movement at the onset or offset of locomotion or head rotation (*Figure 3—figure supplement 2*).

## Correlated activity in the cerebellar-cortical circuit

To understand the relationship between cerebellum and ACC modulation during social interaction and non-social epochs, we performed correlation analyses between the different cell types in the two regions. Previous reports indicate the cerebellum indirectly connects to the ACC via the ventral medial thalamus (*Figure 4A*), but it is not clear whether this indirect connection drives ACC firing during social bouts (*Badura et al., 2018*; *Kelly et al., 2020*). We, therefore, calculated the distribution of correlation coefficients of activity during both the social interaction bouts as well as during the non-social (off-bouts) periods for different PC-ACC and DN-ACC cell pairs depending on their modulation by social interaction (*Figure 4B and C*). Overall, as expected, cerebellar PC and DN neurons and ACC neurons that were similarly modulated by social interaction showed higher correlation values compared to cell pairs that were modulated in opposite directions (*Figure 4D–G*). On the other hand, nsPC or nsDN activity correlation distributions with all ACC groups were similar (*Figure 4—figure supplement 1I,J* & 2). These patterns were largely consistent when the proportion of neurons significantly correlated was calculated between the differently modulated cell types in the cerebellum and ACC (*Figure 4I–K*). Interestingly, these differences almost entirely disappeared when only non-social epochs were analyzed (*Figure 4D–G* insets). Pairwise comparisons of correlation coefficients recorded in the same socially modulated cell pairs with social bouts included and excluded supported our findings (*Figure 4—figure supplement 1* A-H). These findings suggest that correlated activity is largely driven by social interactions and not by the intrinsic connectivity of these neuronal groups.

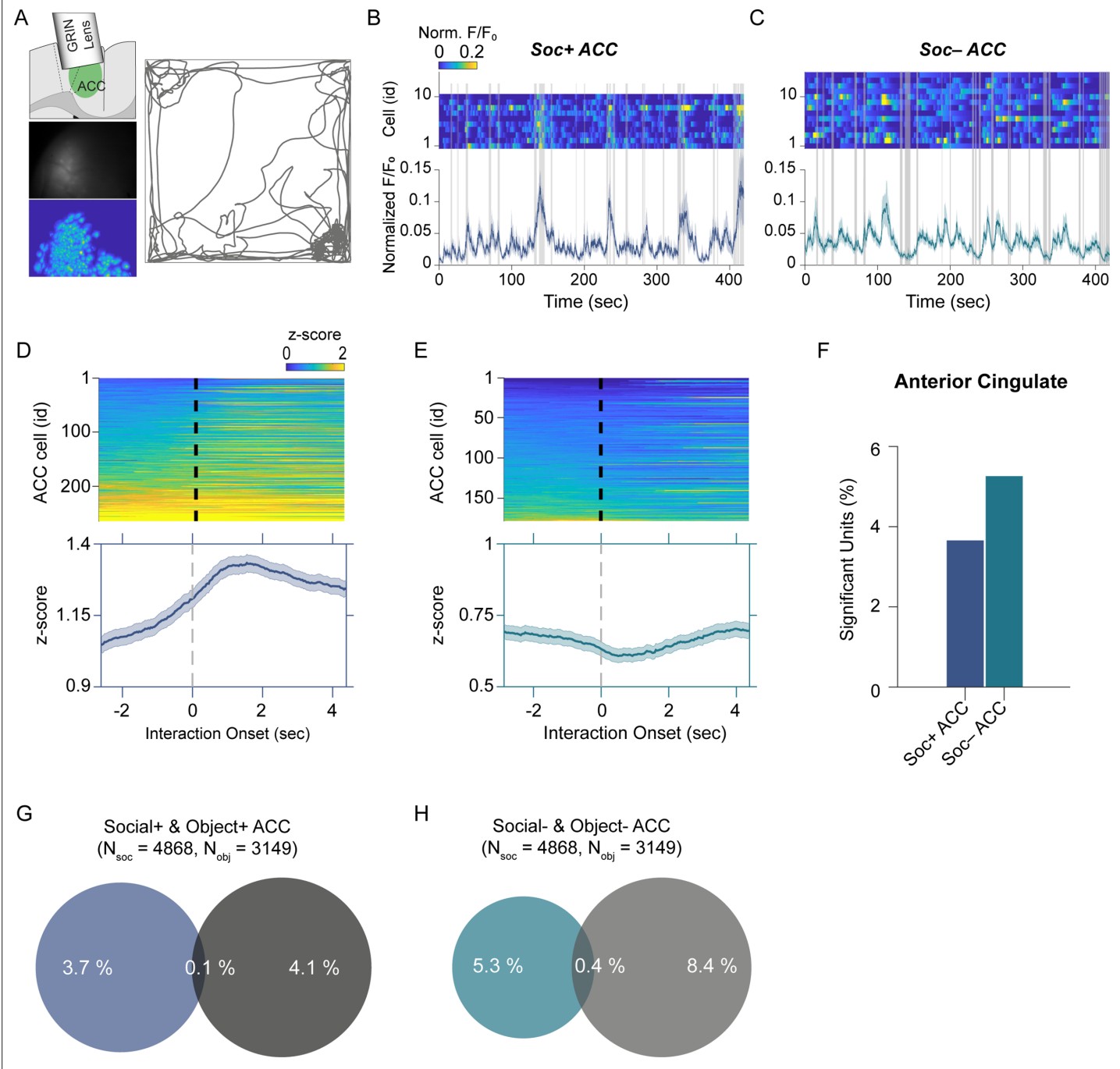

**Figure 3.** Anterior cingulate cortex (ACC) neuron activity patterns during social behavior. (**A**) Illustration of GRIN lens implant location (top left). Raw image from ACC calcium imaging recording (*mid left*). Segmented and averaged activity heatmap from the recording shown above (*bottom left*). Example of raw behavioral trajectory of subject mouse (*right*). (**B, C**) Heatmap depicting the calcium activity of 10 Soc+ ACC neurons (**B**) and 15 Soc– ACC neurons (**C**) during a single social interaction session. Social interaction bouts are shown as gray bars. The average calcium activity is shown below each heatmap. (**D, E**) Top: Z-scored Soc+ (**D**) and Soc– (**E**) ACC neurons calcium activity for all neurons during all social interaction sessions. The onset of social interaction is marked as a dashed black line. Bottom: Mean of Z-scored activity shown for the heatmaps above for Soc+ (**D**) and Soc– (**E**) neurons. (**F**) Percentage of units showing significant modulation by social interaction. (**G**) Overlap of Soc+ and Obj + ACC neuronal populations. (**H**) Overlap of Soc– and Obj– ACC neuronal populations.

The online version of this article includes the following figure supplement(s) for figure 3:

**Figure supplement 1.** Anterior cingulate cortex (ACC) neuron activity patterns during object interaction.

**Figure supplement 2.** Socially excited and inhibited anterior cingulate cortex (ACC) neurons are not related to the onset or offset of locomotion bouts or onset angular head rotation.

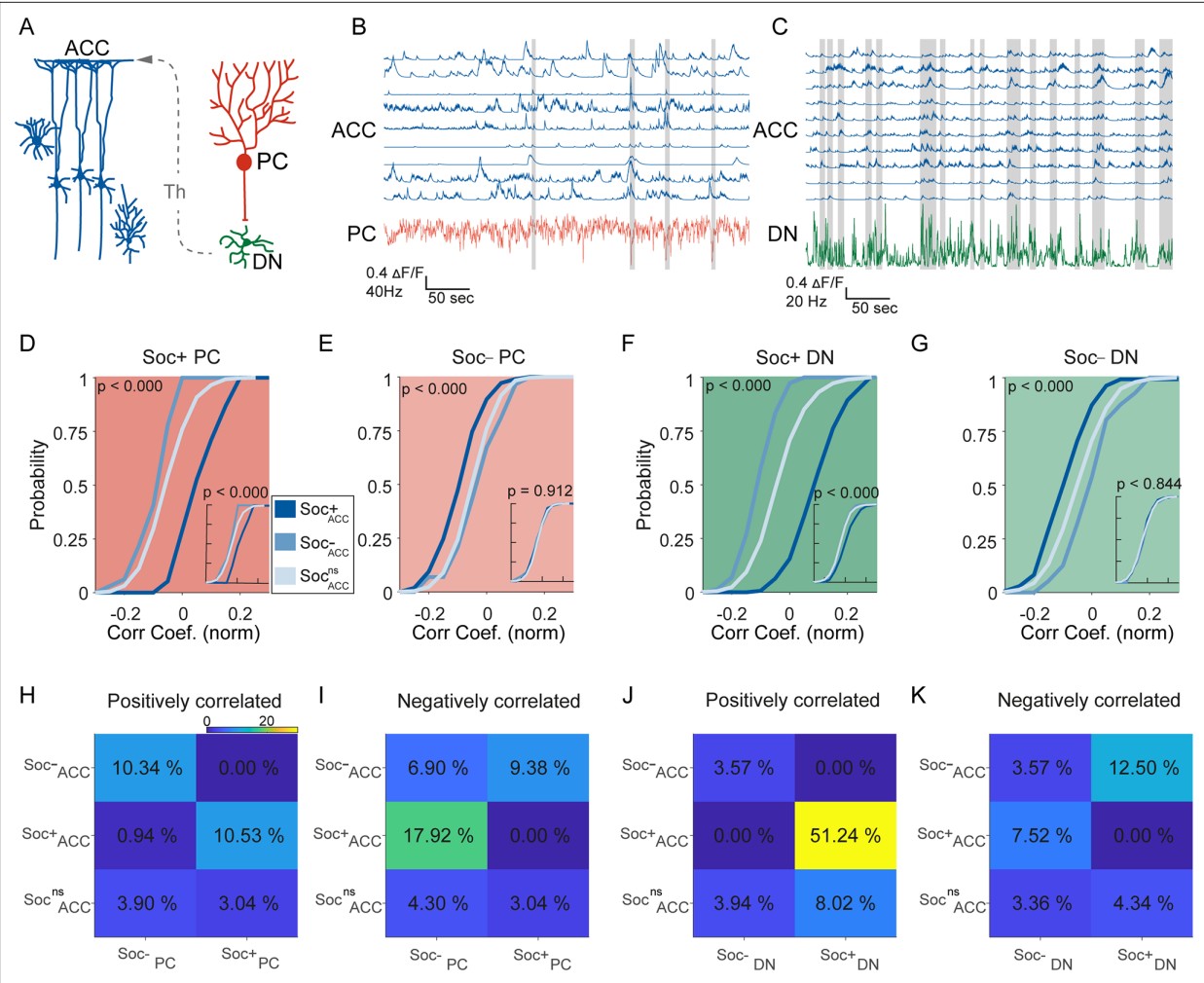

**Figure 4.** Correlated activity in the cerebellar-cortical circuit during social behavior. (**A**) Illustration of the cerebellar-cortical circuit. Purkinje cells (PCs) provide converging inhibition to dentate nucleus (DNs) that excite thalamic neurons. Thalamic neurons excite anterior cingulate cortex (ACC) neurons. (**B**) Simultaneously recorded calcium traces from nine ACC neurons and the electrophysiologically recorded firing rate of a single PC (red) during a social interaction epoch. Social interaction bouts are shown as gray bars. (**C**) Simultaneously recorded calcium traces from 10 ACC neurons (blue) and the electrophysiologically recorded firing rate of a single DN neuron during a social interaction epoch. Social interaction bouts are shown as gray bars. (**D–G**) Cumulative histogram of the distribution of the correlation coefficients for the activity of (**D**) Soc+ PCs, (**E**) Soc– PCs, (**F**) Soc+ DNs, or (**G**) Soc– DNs with Soc+ (dark blue), Soc– (light blue), and Soc$^{ns}$ (blue gray) ACCs. Insets: cumulative histogram of the activity of each set of neurons calculated during periods when the mouse was not engaged in social interaction. (**H, I, J, K**) Correlation matrix showing the percentage of cell pairs showing significant positive (**H, J**) or negative (**I, K**) correlations in activity between Soc+ and Soc– PCs (**H, I**) or Soc+ and Soc– DNs (**J, K**) with Soc+, Soc–, and Soc$^{ns}$ ACC neurons. The color of the squares represents the proportion of neurons correlated.

The online version of this article includes the following figure supplement(s) for figure 4:

**Figure supplement 1.** Relationship of significant social cell pair correlation coefficients during social on- and off-bouts.

**Figure supplement 2.** Purkinje cells and dentate nucleus neurons are not correlated to anterior cingulate cortex (ACC) neuron activity in socially neutral neurons and during off-bout periods.

## Discussion

Social interaction between animals involves a complex set of behaviors that involves the integration of multimodal information and communication between several brain areas. Here, we provide a new perspective on the physiology of the cerebellum during social interaction and how it communicates with downstream brain regions. To faithfully record this synchronized activity, we developed a new device (E-Scope) to record two remote but functionally connected regions of the brain via calcium imaging and electrophysiology. We show that the majority of RCrus I PCs modulated by social interaction decrease their SS firing activity. Conversely, most modulated DN neurons increased their firing

rate during social interaction, consistent with disinhibition of DN firing by simultaneous decreases in PC SS firing rate. In vivo calcium imaging during social interaction showed that less than 10 % of ACC neurons were modulated by social interaction with the majority of neurons being inhibited. As expected, cerebellar and ACC neurons similarly modulated by social interaction showed higher correlations in their firing rate than cerebellar and ACC neurons that were modulated in opposite directions. These differences in correlated firing largely disappeared when only non-social bouts were analyzed, indicating that correlations were dependent on social behavior and not on the intrinsic indirect connectivity of the two brain regions.

## The E-Scope: An integrated device for simultaneous calcium imaging and electrophysiology

The development of the miniature microscope (*Ghosh et al., 2011*) in conjunction with the invention of highly sensitive and robust genetically encoded calcium indicators (*Chen et al., 2013*; *Nakai et al., 2001*) has allowed the recording of activity in large populations of neurons in freely behaving mice. Open-source miniaturized microscopes (*Aharoni et al., 2019a*; *Cai et al., 2016*) have expanded access to this tool and encouraged new developments in Miniscope technology including the development of wire-free (*Shuman et al., 2020*) and large field of view miniaturized microscopes (*Guo et al., 2021*). In recent years, a miniaturized microscope called the NINscope was developed for multi-region recording of calcium signals (*de Groot et al., 2020*), allowing simultaneous calcium imaging complex spike activity in the cerebellum and motor cortical activity during free behavior. However, calcium imaging in PCs provides only a partial picture of their activity, as it reports only the rate of CSs, but not the dynamic and persistent SS firing that is characteristic of these fast-spiking neurons. Here, we introduce the E-Scope, which can synchronously acquire calcium imaging data from one area and electrophysiological data from another to allow simultaneous examination and integration of electrical, calcium, and behavioral data streams. The E-Scope also provides method flexibility, allowing simultaneous recordings with areas where implanting a lens may be too invasive, such as deep brain regions like the DN where lens implantation would require the removal of its input layer in the cerebellar cortex. Furthermore, the E-Scope can be used to record local field potentials from any brain region, regardless of depth, simultaneously with calcium imaging. This approach therefore, provides a unique tool to identify neurons activated during different oscillations critical for memory consolidation and cognition such as ripple and theta oscillations in the hippocampus or spindle oscillations in the cortex. Additionally, the integration of data streams and Miniscope power into a thin single coax cable is less limiting to the physical movement of the animal than multiple wires, enabling better interpretation of physiological correlates of behavior. Moreover, the E-Scope can be used with multiple different extracellular electrophysiological recording techniques including silicon microprobes as shown here, as well as tetrode arrays (*Howe and Blair, 2022*) or flexible electrode arrays (*Chung et al., 2019*). Several approaches now exist for linking different types of electrodes to GRIN lenses for simultaneous electrophysiological recordings and calcium imaging from the same brain region (*Cobar et al., 2022*; *McCullough et al., 2022*; *Wu et al., 2021*). The E-Scope would be able to couple with these electrode systems/imaging systems as well.

## Electrophysiological recordings demonstrate reciprocal social modulation between RCrus I PCs and DNs

Lesion and imaging studies in humans have suggested that the cerebellum's role extends beyond motor control and that the cerebellum may be important for social behavior (*Schmahmann and Sherman, 1998*; *Sokolov et al., 2017*). In fact, cerebellar abnormalities are one of the most replicated anatomical changes in brains of individuals with autism (*Fatemi et al., 2012*). A causal role for the cerebellum in social behavior was further solidified when specific genetic or activity modulations of RCrus I had dramatic effects on social behavior (*Badura et al., 2018*; *Kelly et al., 2020*). Yet, how distinct cerebellar cell types in this brain region are modulated by social interaction is not known as it requires direct electrophysiological recordings from cerebellar neurons in freely behaving animals. Our study is the first to demonstrate physiological cerebellar PC and DN neuron activity during social interaction in freely behaving animals. We find that most socially modulated PC neurons in RCrus I are inhibited. The mechanisms driving prolonged inhibition of PC firing during social interaction is still not clear. This inhibition lasts far longer than would be expected post-complex spike pauses in activity.

One potential mechanism could be coordinated decreases in cerebellar granule cell activity during social interaction dynamically decreasing excitatory input to PCs and, therefore, reducing PC SS firing. Alternatively, inhibition of PC firing rate may be due to an increase in inhibitory versus excitatory input from molecular layer interneurons (MLIs) versus granule cells, respectively. A process that could be driven by LTP and LTD of the corresponding synapse during learning. Previous in vivo studies have shown that sensory stimulation can increase MLI activity (*Jörntell and Ekerot, 2003*) and, therefore, social sensory input such as olfactory stimuli could potentially increase MLI firing. This potential mechanism of PC inhibition is supported by experiments where suppression of Crus I MLIs causes changes in social preference (*Badura et al., 2018*), suggesting that MLI output may play a prominent role in modulating PC output and, therefore, DN output during social behavior. Future recordings from cerebellar granule neurons or MLI neurons during social interaction will be necessary to answer these questions.

Bidirectional modulation of activity in the cerebellum is a well-known phenomenon (*De Zeeuw, 2021*). We found bidirectional modulation of PC firing during social interaction with the majority of PCs inhibited by social interaction and a minority excited. Consistent with the inhibitory role of PCs onto DNs, we find this ratio to be inverted in DNs with the majority of modulated DNs excited and only minority inhibited. As PC's receives inputs from parallel fibers, which convey proprioceptive, peripheral sensory and motor-related information (*Gao et al., 2018*), we performed analysis to demonstrate that our social activity modulations were not simply related to motor behavior. Movement of the animal at the onset, offset, or changes in locomotion direction could not explain the modulations of activity we recorded in the cerebellum, suggesting that activity modulations were driven specifically by social interaction. In addition, there was little overlap in the neurons modulated by social interaction and by object interactions. This indicates that RCrus I activity is responsive to elements of social interaction itself, although the specific information encoded is still to be ascertained.

## Correlated firing between PCs, DNs, and ACC neurons

An indirect connection links RCrus I cerebellum to ACC via the thalamus (*Badura et al., 2018*). DN neurons project to the ventromedial thalamus, which in turn projects to the ACC (*de Lima et al., 2022*; *Kuramoto et al., 2015*). As expected, we find that correlations between PCs or DNs with ACC neurons are greater when these neurons are similarly modulated by social interaction. However, these differences in correlations disappear when only non-social epochs are analyzed, suggesting that spontaneous activity during non-social behavior and the intrinsic connectivity of the two regions is not sufficient to impose differences in correlations without social interaction. These results suggest that communication between the cerebellum and ACC is coordinating social behavior. However, future studies are necessary to understand more fully what the specific behaviorally relevant signals that are being output from the cerebellar circuit to the ventromedial thalamic neurons and then to the ACC.

Our findings support the theory that the cerebellum plays a role in social behavior. Moreover, the E-Scope provides an important tool for examining the activity of two distant brain regions during behavior. We have further uncovered how cerebellar activity indirectly impacts ACC neurons output during social behavior and non-social behavior. These physiological insights into cerebellar and ACC communication during social behavior may eventually provide ways to finely tune cerebellar activity in autism models and eventually in humans to augment and improve the quality of social interactions.

## Methods
### E-Scope parts and assembly

A two-shank, 64 channel silicon probe (64 H with 32 channels activated) was used that was wire bonded and epoxied onto a probe PCB (PCB: Hughes Circuits, assembly: IDAX Microelectronics). The probe PCB that incorporated the ground wire was connected to a slimstack connector (Molex) via a flat flex cable (*Figure 1—figure supplement 1A*). The male slimstack connector on the probe was then connected to the female slimstack connector Ephys PCB (*Figure 1—figure supplement 1B*). This Ephys PCB incorporated a 32-channel electrophysiology-integrated chip (Intan, RHD2132) and a 32-bit ARM-Cortex microcontroller unit (MCU) (Atmel; ATSAMS70N21A). This MCU configures the Intan electrophysiology chip and polls it using direct memory access (DMA) over a single-ended serial peripheral interface (SPI) at up to 20 kSps per channel for new electrophysiology data across its 32

channels. The MCU then repackages the serialized electrophysiology data into 512-bit-long packets, containing one 16-bit sample for each of the 32 channels. These data packets are then clocked out by the E-Scope's image sensor pixel clock using the MCU's synchronous serial controller (SSC) and DMA. The resulting timing and structure of the electrophysiology data allows for this data to be further serialized with the image sensor data due to all data now sharing the same acquisition clock, namely the image sensor pixel clock. The Ephys PCB was connected through a six-wire cable assembly soldered between the Ephys PCB and the V3 CMOS PCB (*Figure 1*, *Figure 1A—figure supplement 1C*), allowing integration and synchronization of electrophysiological and imaging data. The wires connecting the Ephys PCB to the CMOS PCB were fixed in place using hot melt adhesive to protect the wires during behavior. A single, flexible, 1.1 mm diameter, coaxial cable (Cooner Wire, CW2040-3650SR) connected the integrated E-Scope with all off-board power and DAQ hardware. This single cable supplies the E-Scope with power, carries low-bandwidth bidirectional communication data, and carries unidirectional high-bandwidth synchronized imaging and electrophysiology data to an open-source UCLA Miniscope DAQ connected by a coaxial SMA connector (*Figure 1—figure supplement 1D*). Miniscope imaging data acquired by the Miniscope DAQ was sent over USB using a USB video device (UVC) protocol while electrophysiology data carried over the coaxial cable was routed from the UCLA Miniscope DAQ to an Intan DAQ (RHD 2000 evaluation board) via an intermediary circuit used to mimic a standard 32 channel Intan headstage (*Figure 1—figure supplement 1D–G*). This intermediary circuit and PCB has the same MCU that is on the Ephys PCB and effectively undoes the timing and packeting steps implemented on the E-Scope as well as emulates the registers of the Intan electrophysiology amplifier chip and converts single-ended SPI to LVDS SPI (*Figure 1—figure supplement 1E*). These two acquisition systems were then synchronized together using the 'frame sync' output signal from the Miniscope DAQ as a digital input to the Intan DAQ. This served as a common low-voltage transistor-transistor logic (LVTTL) signal for each Miniscope frame acquired by the Miniscope DAQ and enabled alignment of electrophysiology and image data offline. The components necessary for building an E-Scope are listed in *Table 1*.

## Animals

6–8 week-old C57BL/6 J young adult male mice at the time of initial surgery were used for in vivo E-Scope calcium imaging and electrophysiology experiments. All mice were acquired from Jackson Laboratories and group-housed three per cage on a 12 hr light-dark cycle. For all behavioral experiments, aged-matched novel animals were used. All experimental protocols were approved by the Chancellor's Animal Research Committee of the University of California, Los Angeles in accordance with the U.S. National Institutes of Health (NIH) guidelines.

## Viral vectors

Stereotaxic injections for E-Scope experiments were done using a stereotaxic frame (David Kopf Instruments) and a Nanoject II microinjector (Drummond Scientific). For Miniscope calcium imaging experiments, AAV1. Syn.GCaMP6f.WPRE.SV40 virus (titer 4.5 × 1013 GC/mL; Penn Vector Core) was injected.

## In vivo E-Scope surgeries

Mice were anesthetized with 1.5–2.0% isoflurane and placed into a stereotaxic apparatus (David Kopf Instruments) for all surgeries. Once the depth of anesthesia was confirmed by the absence of reflex, the mouse was moved to a heat pad (Harvard Apparatus) on the stereotaxic frame. The ear bars were firmly fixed onto the skull of the mouse. Eye ointment was applied to protect the eyes from drying. Hair was shaved off and incision sites were sterilized with multiple iterations of ethanol and beta-iodine before going into surgery.

The mouse skull was calibrated so that the bregma and lambda were aligned and on the same plane. AAV1. Syn.GCaMP6f.WPRE.SV40 virus was injected in the ACC (Anterioposterior: 0.9 mm, Mediolateral: 0.2 mm, Dorsoventral: –1.3 mm relative to bregma; 300 nL) using a Nanoject II microinjector (Drummond Scientific) at 60 nl·min$^{-1}$. The location of the DN (Anteroposterior: –5.8 mm, Mediolateral: –2.25 mm relative to bregma) or RCrus I (Anteroposterior: –6.5 mm, Mediolateral: –2.5 mm relative to bregma) was labeled for later access. The incision site was sutured using silk suture threads

**Table 1.** List of components for E-Scope Assembly.

| Component | Quantity | Vendor | Part # | Weblink |
|---|---|---|---|---|
| **Miniscope V3 Parts** | | | | https://miniscope.org |
| *Body* | | | | |
| Main body of scope. Black Delrin | 1 | N/A | MS_MainBody v3.2 | |
| Filter cover of scope. Black Delrin | 1 | N/A | MS_FilterCover v3 | |
| CMOS imaging sensor mount. Black Delrin | 1 | N/A | MS_FocusSlider v3.2 | |
| Baseplate. Aluminum. | 1 | N/A | MS_Baseplate v3 | |
| Cap to protect implanted GRIN lens. Black Delrin | 1 | N/A | MS_Cap v3 | https://github.com/daharoni/Miniscope_Machined_Parts |
| *Optics* | | | | |
| 5 mm Dia. × 20 mm FL, MgF2 Coated, Achromatic Doublet Lens | 1 | Edmund Optics | 45–408 | http://www.edmundoptics.com/optics/optical-lenses/achromatic-lenses/mgf2-coated-achromatic-lenses/45408/ |
| 5 mm Dia. × 15 mm FL, MgF2 Coated, Achromatic Doublet Lens | 1 | Edmund Optics | 45–207 | http://www.edmundoptics.com/optics/optical-lenses/achromatic-lenses/mgf2-coated-achromatic-lenses/45207/ |
| 5 mm Dia. × 12.5 mm FL, MgF2 Coated, Achromatic Doublet Lens | 1 | Edmund Optics | 49–923 | http://www.edmundoptics.com/optics/optical-lenses/achromatic-lenses/mgf2-coated-achromatic-lenses/49923/ |
| 5 mm Dia. × 10 mm FL, MgF2 Coated, Achromatic Doublet Lens | 1 | Edmund Optics | 45–206 | http://www.edmundoptics.com/optics/optical-lenses/achromatic-lenses/mgf2-coated-achromatic-lenses/45206/ |
| 5 mm Dia. × 7.5 mm FL, MgF2 Coated, Achromatic Doublet Lens | 1 | Edmund Optics | 45–407 | http://www.edmundoptics.com/optics/optical-lenses/achromatic-lenses/mgf2-coated-achromatic-lenses/45407/ |
| 3.0 mm Diameter, N-BK7 Half-Ball Lens | 1 | Edmund Optics | 47–269 | http://www.edmundoptics.com/optics/optical-lenses/ball-condenser-lenses/n-bk7-half-ball-lenses/47269/ |
| Diced excitation filter, 3.5mm × 4 mm × 1 mm | 1 | Chroma | ET470/40 x | https://www.chroma.com/products/parts/et470-40x |
| Diced dichroic mirror, 6mm × 4mm × 1 mm | 1 | Chroma | T495lpxr | https://www.chroma.com/products/parts/t495lpxr |
| Diced emission filter, 4mm × 4mm × 1 mm | 1 | Chroma | ET525/50 m | https://www.chroma.com/products/parts/et525-50m |
| *Electrical* | | | | |
| Excitation LED, LED LUXEON REBEL BLUE SMD | | Digikey | LXML-PB01-0030 | http://www.digikey.com/product-detail/en/LXML-PB01-0030/1416-1028-1-ND/3961133 |
| *Coaxial Cable* | | | | |
| 50 ohm coax silicone rubber jacketed cable | | Cooner Wire | CW2040-3650SR | https://www.coonerwire.com/mini-coax/ |

*Table 1 continued on next page*

*Table 1 continued*

| Component | Quantity | Vendor | Part # | Weblink |
|---|---|---|---|---|
| *Printed Circuit Boards (PCB)* | | | | |
| 4 layer, 0.031″ CMOS imaging sensor PCB | 1 | N/A | N/A | https://github.com/daharoni/Miniscope_CMOS_Imaging_Sensor_PCB |
| 2 layer, 0.031″ Excitation LED PCB | 1 | N/A | N/A | https://github.com/daharoni/Miniscope_Machined_Parts/tree/master/Extra%20Components |
| 2 layer, 0.062″ Coax to SMA PCB | 1 | OSH Park | N/A | |
| *Electrophysiology Parts* | | | | |
| *Probe* | | | | |
| 64 channel silicon probe (with 32 channels wirebonded) | 1 | Sotiris Lab | 64 H | https://github.com/sotmasman/Silicon-microprobes https://masmanidislab.neurobio.ucla.edu/technology.html |
| *Printed Circuit Boards (PCB)* | | | | |
| Ephys PBC | 1 | N/A | N/A | https://github.com/Aharoni-Lab/Ephys-Miniscope |
| *Miscellaneous Hardware* | | | | |
| M1 thread-forming screws | 4 | McMaster-Carr | 96817a704 | https://www.mcmaster.com/tappingscrews/screw-size~m1/ |
| Set Screw 18–8 Stainless Steel Cup Point Set Screw, 0–80 Thread, 3/16″ Long | 2 | McMaster-Carr | 92311 A054 | https://www.mcmaster.com/92311A054/ |
| (Black) 36 Gauge Ultra-Flexible Miniature High-Temperature Wire | 1 | McMaster-Carr | 9564T1 | |
| (Green) 36 Gauge Ultra-Flexible Miniature High-Temperature Wire | 1 | McMaster-Carr | 9564T1 | |
| (Red) 36 Gauge Ultra-Flexible Miniature High-Temperature Wire | 1 | McMaster-Carr | 9564T1 | |
| (White) 36 Gauge Ultra-Flexible Miniature High-Temperature Wire | 1 | McMaster-Carr | 9564T1 | https://www.mcmaster.com/wire/wire-gauge~36/ |
| 1/16″ diameter × 1/32″ thick. Axially Magnetized | 6 | K&J Magnetics, Inc. | D101-N52 | https://www.kjmagnetics.com/proddetail.asp?prod=D101-N52 |
| *Data Acquisition Devices (DAQs)* | | | | |
| UCLA Miniscope DAQ | 1 | Labmaker | DAQ-Imaging | https://www.labmaker.org/collections/miniscope-v3-2/products/data-aquistion-system-daq |
| SSC-2-Intan-LVDS PCB | 1 | N/A | N/A | https://github.com/Aharoni-Lab/Ephys-Miniscope |
| Intan DAQ | 1 | Intan Technologies | RHD 2000 evaluation board | https://intantech.com/RHD_USB_interface_board.html |

*Table 1 continued on next page*

*Table 1 continued*

| Component | Quantity | Vendor | Part # | Weblink |
|---|---|---|---|---|
| **Software** | | | | |
| | | | | https://github.com/Aharoni-Lab/Miniscope-DAQ-Cypress-firmware (*Aharoni and Klumpp, 2023*) |
| Aharoni-Lab Miniscope-DAQ-QTSoftware | 1 | N/A | N/A | https://github.com/daharoni/Miniscope_DAQ_Software (*Aharoni et al., 2019b*) |
| Intan RHX software | 1 | Intan Technologies | RHX | https://intantech.com/RHX_software.html (*Intan Technologies, 2024*) |

(#18020–00, Fine Science Tools). Mice were given carprofen analgesic (5 mg·kg$^{-1}$) for 3 days and amoxicillin antibiotic (0.25 mg·ml-1) through ad libitum water supply for 3 days.

After a week of recovery, mice underwent relay GRIN lens (Inscopix; PN 130–000143) implant surgery. After sterilizing the scalp, an incision was made to expose the skull. A skull screw and a ground pin were fastened to the skull, followed by a craniotomy with a diameter of 1 mm was made above the virus injection site. The tissue above the targeted implantation site was carefully aspirated using a 30-gauge blunt needle. Buffered ACSF was constantly applied throughout the aspiration to prevent the tissue from drying. Aspiration ceased after the target depth (0.7 mm) had been reached and full termination of bleeding. Here, a 1 mm relay GRIN lens (1 mm diameter, 4 mm length, Inscopix) was stereotaxically lowered and implanted at a 10 degree angle to the target site (–0.7 mm dorsoventral from the skull surface relative to the most posterior point of the craniotomy). Cyanoacrylate adhesive (Vetbond, 3 M) and dental cement (Ortho-Jet, Lang Dental) were applied to fix the lens in place as well as to cover the exposed skull. Kwik-Sil was used to protect the protruding relay GRIN lens. Carprofen (5 mg·kg$^{-1}$) and dexamethasone (0.2 mg·kg-1) were administered subcutaneously during and after surgery along with amoxicillin (0.25 mg·ml-1) in the drinking water for 3 days.

Mice were anesthetized again 2 weeks subsequently, and a baseplate locked to a Miniscope containing an objective GRIN lens (2 mm diameter, 4.79 mm length, 0.25 pitch, 0.50 numerical aperture, Grintech) was placed above the relay lens to search for the optimal in-focus cells in the field of view. Once field of view was obtained, the baseplate was cemented in place, and the Miniscope was unlocked and detached from the baseplate. A plastic cap was attached on the top of the baseplate to prevent debris build-up.

Mice were anesthetized for the fourth time for silicon multichannel electrode probe implantation, which were manufactured using the same process previously reported by *Yang et al., 2020*. The 2-shank 32 channel silicon probe PCB was screwed to a holder and coated with DiI (#C7000, Thermo Fisher Scientific) prior to insertion. A 1 mm diameter craniotomy was made on the previously marked DN or RCrus I locations. A micro-incision was made in the dura for the probe to go in. The probe was slowly lowered to the target locations (DN: –2.0 mm or RCrus I: 200–250 μm) where optimal signal-to-noise ratio was obtained and waited for the probe to settle in the brain for more than 30 min. Once location was set, the probe was dental cemented into place. Mice were given carprofen and dexamethasone post-surgery for 3 days along with amoxicillin in drinking water.

## Habituation

Subject mice were habituated on a dummy Miniscope which had the equal amount of weight as the E-Scope for 7–10 days prior to multichannel silicon probe implant. Weights (0.55 g/piece; DILB8P-223TLF, Digi-Key Electronics) were increased 0.55 g every 2 days. Habituation was halted once the mice were active enough to carry the weight of the dummy Miniscope (4.5 g). All novel target mice were pre-habituated 15 min each day, for 3 days to the arena prior to experiments. Before any of our social interaction behavioral experiments, aggressive or agitated mice were removed after assessing their behavior in the arena while habituation.

## Social behavior and data analysis

Behavioral experiments were held in a low light (20–50 LUX) environment with white noise (50–3 kHz). Subject mice were placed in a 48 × 48 cm arena for 1 min prior to being introduced to either a novel

object or novel target mouse. For all trials, all mice or objects were introduced for the first time. Social or object interactions were defined by the proximity between the subject mouse and novel target mouse or object (2 cm from the body, head, or base of tail). A novel object or a novel target mouse was initially placed in the middle of the arena and introduced for 7 min. Between all recording sessions, the arena was cleaned with 70% ethanol. Behavior sessions were recorded using a webcam (C920, Logitech) connected to the host computer using the Miniscope software.

Social interaction behavioral analyses were done by utilizing a custom Python script to trace the centroid of the body outline of both the subject and novel target mouse or object and tracking their proximity to each other. Either nose-to-nose, nose-to-rear, or nose-to-body were considered as social interactions. We traced the LED on the Miniscope CMOS PCB located on the head of the subject mouse to calculate the speed and head angular speed. Statistical analysis was done on Prism (GraphPad software).

## In vivo E-Scope recording and data analysis

Recordings proceeded 2 days post probe implant. The body of the E-Scope was first connected to the Miniscope-DAQ (*Figure 1—figure supplement 1*). Ground wires were set in place before powering the DAQ. The Miniscope-DAQ as well as the Intan DAQ were connected to the host computer. The UCLA Miniscope and Intan electrophysiology data acquisition programs were opened and run to confirm the imaging region of interest and electrophysiological signal quality. Next, the electrophysiological amplifier PCB was connected to the Miniscope via 6-wire cable assembly. The Miniscope portion of the E-Scope was fixed onto the baseplate of the mouse's head, followed by the connection of the electrophysiology amplifier PCB to the implanted probe in the mouse.

For cerebellar electrophysiological data analysis, spike sorting was performed by Kilosort 2.5 (*Pachitariu et al., 2016*) and P-sort (*Sedaghat-Nejad et al., 2021*). Isolated units were further manually curated in phy2 on other single unit activity criteria in addition to merging and splitting highly similar and mixture units, respectively. Cross-correlogram between simple and complex spikes were constructed for the spike trains of the putative Purkinje cell units that revealed signature complex spike induced simple spike pauses (*Figure 1G*). Further analyses were done using custom script in MATLAB (MathWorks,2018).

ACC calcium imaging analysis was done using the Miniscope Analysis pipeline (*Etter et al., 2020*). Videos went through NoRMCorre for motion correction (*Pnevmatikakis and Giovannucci, 2017*), then videos acquired during a session were concatenated to extract spatial components and calcium traces of individual neurons using CNMF-E (*Zhou et al., 2018*). Similar to electrophysiology analysis, the calcium traces were normalized, and Z-score values were used to test for the social or object interaction- and movement-initiated activity.

To synchronize electrophysiology and calcium imaging data, we binned unit's spike trains using 33ms bins to match with calcium traces recorded at ~30 Hz. Spike trains were further convolved with the Gaussian kernel with sigma of 100 ms. The spike rate maps were Z-scored for further analysis. Correlation analysis was computed using Pearson correlation (*Figure 4*), which was tested for significance by comparing correlations to those obtained from 1000 randomly circularly shifted calcium trace data. Cross-correlation was computed between spike times and deconvolved signals.

## Histology

Post-hoc histology was performed to confirm the location of the silicon probe. Both silicon probe shanks were coated with DiI (#C7000, Thermo Fisher Scientific). After the recordings, mice were deeply anesthetized with isoflurane then cardiac perfused with phosphate-buffered saline (PBS) followed by 4% Paraformaldehyde (PFA; Sigma-Aldrich). Whole brains were carefully dissected and post-fixed into 4% PFA overnight and sliced using a vibratome (Leica VT1200). Coronal serial sections were made at a thickness of 70 μm in a 4 °C PBS solution, mounted on a slide glass, and cover slipped with VECTA-SHIELD (H-1400, Vector Laboratories). We used the Paxinos Mouse Brain Atlas (*Franklin and Paxinos, 2008*) for nomenclature of brain regions. Images were taken using a widefield microscope (Apotome, Ziess). Contrast, brightness, and pseudocolor were adjusted in Image J (*Schneider et al., 2012*). Images were tilted to align to illustration based on the atlas.

## Statistics

Statistical analyses were conducted using custom written scripts in MATLAB (MathWorks, 2018) and Prism (GraphPad software). No statistical methods were used to determine sample size. Sessions with no object interaction were excluded for electrophysiology and calcium imaging data analysis. Kolmogorov Smirnov normality test was applied before using Wilcoxon rank sum test for hypothesis testing. Non-parametric Kruskal-Wallis H test was used in some instances to cross validate significance for some of the figure supplements. Otherwise, the t-test was used. p-value indicates one-tailed values and significance level of 0.01 was used unless stated otherwise.

## E-Scope availability

All design files, software, parts list, assembly details and tutorials to build the E-Scope are available at http://miniscope.org/index.php/Main_Page and https://github.com/Aharoni-Lab/Ephys-Miniscope.

## Acknowledgements

We thank Dr. Tom Otis for supporting this project. We are very thankful to Ebrahim Feghhi, Alejandro Hipolito, Jason Kwon, and Marcus Min for handling animals and the technical support. This work was supported by the VA Merit Award BX005202 (to PG), NSF NeuroNex Award 1707408 (to PG, DA, SM, and TB), NIH P50HD103577 (to PG and DA), NIH U01NS122124 (to PG and DA), NIH R01NS090930 (to PG), and NIH 1R61NS119708 (to PM).

## Additional information

### Funding

| Funder | Grant reference number | Author |
| --- | --- | --- |
| Veterans Affairs | Merit Award BX005202 | Peyman Golshani |
| National Science Foundation | NeuroNex Award 1707408 | Hugh T Blair |
| National Institutes of Health | P50HD103577 | Daniel Aharoni |
| National Institutes of Health | U01NS122124 | Daniel Aharoni |
| National Institutes of Health | R01NS090930 | Peyman Golshani |
| National Institutes of Health | 1R61NS119708 | Paul J Mathews |

The funders had no role in study design, data collection and interpretation, or the decision to submit the work for publication.

### Author contributions

Sung Won Hur, Conceptualization, Resources, Data curation, Software, Formal analysis, Validation, Investigation, Visualization, Methodology, Writing – original draft, Writing – review and editing; Karen Safaryan, Conceptualization, Resources, Data curation, Software, Formal analysis, Validation, Investigation, Visualization, Writing – original draft; Long Yang, Methodology; Hugh T Blair, Investigation, Methodology, Writing – review and editing; Sotiris C Masmanidis, Methodology, Writing – review and editing; Paul J Mathews, Conceptualization, Formal analysis, Supervision, Investigation, Writing – original draft, Project administration, Writing – review and editing; Daniel Aharoni, Conceptualization, Resources, Software, Supervision, Funding acquisition, Investigation, Methodology, Writing – review and editing; Peyman Golshani, Conceptualization, Resources, Formal analysis, Supervision, Funding acquisition, Investigation, Methodology, Writing – original draft, Project administration, Writing – review and editing

## Author ORCIDs
Sung Won Hur ⓘ https://orcid.org/0000-0003-2699-1386
Sotiris C Masmanidis ⓘ http://orcid.org/0000-0002-8699-3335
Peyman Golshani ⓘ http://orcid.org/0000-0002-5406-7695

## Ethics

This study was performed in strict accordance with the recommendations in the Guide for the Care and Use of Laboratory Animals of the National Institutes of Health. All of the animals were handled according to approved institutional animal care and use committee (IACUC) protocols (#2006-066) of the University of California, Los Angeles.

Reviewer #1 (Public Review): https://doi.org/10.7554/eLife.88439.3.sa1
Reviewer #2 (Public Review): https://doi.org/10.7554/eLife.88439.3.sa2
Reviewer #3 (Public Review): https://doi.org/10.7554/eLife.88439.3.sa3
Author Response https://doi.org/10.7554/eLife.88439.3.sa4

---

# Additional files

## Supplementary files
• MDAR checklist

## Data availability

Data and code for analyzing the data for this study are available at: https://github.com/golshanilab/Escope_Social_Cerebellum_ACC (copy archived at *Hur and Safaryan, 2024*).

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
