## [Editor Report · eLife assessment]

Based on a technological advance which couples onboard calcium imaging with in vivo electrophysiology in freely behaving mice, this **important** work presents data about the modulation of some long range brain activity correlations during social interactions. **Solid** evidence shows that neural activity across cerebellum and cingulate cortex is more correlated during social behaviors than during non-social epochs. This study is of interest for a broad range of neurophysiologists.

---

## [Referee Report · Reviewer #1 (Public Review)]

In this manuscript, the authors describe an improved miniscope they name "E-scope" combining in vivo calcium imaging with electrophysiological recording and use it to examine neural correlates of social interactions with respect to cerebellar and cortical circuits. Through correlations between electrophysiological single units of Purkinje cells and dentate nucleus neurons as well as with calcium signals imaging of neurons from the anterior cingulate cortex, the authors provide correlative data supporting the view that intracerebellar circuits and cerebello-cortical communications take part in the modulation of social behavior. In particular, the electrophysiological dataset reflects the PC-DN connection and strongly suggests its involvement in social interactions. Cross-correlations analyses between PC / DN single units and ACC calcium signals suggest that the recorded cerebellar and cortical structures both take part in the brain networks at play in social behavior.

Comments on revised submission:

While the authors have, to some extent, replied to most of my comments, they seem to have chosen not to respond to the part concerning the different types of social interactions that are not addressed in the manuscript, as also pointed out by reviewer 3. However, I feel that given the scope of the paper, which aims at demonstrating the value of the E-scope new device, this should not preclude the current study from being published.

---

## [Referee Report · Reviewer #2 (Public Review)]

This report by Hur et al. examines simultaneous activity in the cerebellum and anterior cingulate cortex (ACC) to determine how activity in these regions is coordinated during social behavior. To accomplish this, the authors developed a recording device named the E-scope, which combines a head-mounted mini-scope for in vivo Ca2+ imaging with an extracellular recording probe (in the manuscript they use a 32-channel silicon probe). Using the E-scope, the authors find subpopulations of cerebellar neurons with social-interaction-related activity changes. The activity pattern is predominantly decreased firing in PCs and increases in DNs, which is the expected reciprocal relationship between these populations. They also find social-interaction-related activity in the ACC. The authors nicely show the absence of locomotion onset and offset activity in PCs and DNs ruling out that is movement driven. Analysis showed high correlations between cerebellar and ACC populations (namely, Soc+ACC and Soc+DN cells). The finding of correlated activity is interesting because non-motor functions of the cerebellum are relatively little explored. However, the causal relationship is far from established with the methods used, leaving it unclear if these two brain regions are similarly engaged by the behavior or if they form a pathway/loop. Overall, the data are presented clearly, and the manuscript is well written, however the biological insight gained is rather limited.

---

## [Referee Report · Reviewer #3 (Public Review)]

Complex behavior requires complex neural control involving multiple brain regions. The currently available tools to measure neural activity in multiple brain regions in small animals are limited and often involve obligatory head-fixation. The latter, obviously, impacts the behaviors under study. Hur and colleagues present a novel recording device, the E-Scope, that combines optical imaging of fluorescent calcium imaging in one brain region with high-density electrodes in another. Importantly, the E-Scope can be implanted and is, therefore, compatible with usage in freely moving mice. The authors used their new E-Scope to study neural activity during social interactions in mice. They demonstrate the presence of neural correlates of social interaction that happen simultaneously in the cerebellum and the anterior cingulate cortex.

The major accomplishment of this study is the development and introduction of the E-Scope. The evaluation of this part can be short: it works, so the authors succeeded.

The authors managed to reduce the weight of the implant to 4.5 g, which is - given all functionality - quite an accomplishment in my view. However, a mouse weighs between 20 and 40 g, so that an implant of 4.5 g is still quite considerable. It can be expected that this has an impact on the behavior and, possibly, the well-being of the animals. Whether this is the case or not, is not really addressed in this study. The authors suffice with the statement that "Recorded animals made more contact with the other mouse than with the object (Figure 2A), suggesting a normal preference for social contact with the E-Scope attached." A direct comparison between mice before and after implant, or between mice with and without an implant would provide more insight into the putative impact of the E-Scope on (social) behavior.

In Figure 1 D-G, the authors present raw data from the neurophysiological recordings. In panel D, we see events with vastly different amplitudes. It would be very insightful if the authors would describe which events they considered to be action potentials, and which not. Similarly, indicating the detected complex spikes in the raw traces of Figure 1E would provide more insight into the interpretation of the data. Although the authors mention to consider the occurrence of complex spikes and simple spikes, a clear definition of what is considered a single unit recording is lacking. As there is quite a wide range in reported firing rates in Figure 2 - figure supplement 3, more clarity on this aspect would be insightful. Furthermore, in their text, the authors state that the pause in simple spike firing following a complex spike normally lasts until around 40 ms, and for this statement they refer to Figure 1G that shows a pause of less than 10 ms.

The number of Purkinje cells recorded during social interactions is quite low: only 11 cells showed a modulation in their spiking activity (unclear whether in complex spikes, simple spikes or both). During object interaction, only 4 cells showed a significant modulation. Unclear is whether the latter 4 are a subset of the former 11, or whether "social cells" and "object cells" are different categories. Having so few cells, and with these having different types of modulation, the group of cells for each type of modulation is really small, going down to 2 cells/group. The small group sizes complicate the interpretation of the data - in particular also on the analysis of movement-related activity that is now very noisy (Figure 2 - figure supplement 4).

In conclusion, the authors present a novel method to record neural activity with single cell-resolution in two brain regions in freely moving mice. Given the challenges associated with understanding of complex behaviors, this approach can be useful for many neuroscientists. The authors demonstrate the potential of their approach by studying social interactions in mice. Clearly, there are correlations in activity of neurons in the anterior cingulate cortex and the cerebellum related to social interactions. To bring our understanding of these patterns to a higher level, more detailed analyses (and probably also larger group sizes of cerebellar neurons) are required, though.

---

## [Author Response]

The following is the authors’ response to the original reviews.

Positive comments:

We appreciate the positive comments of the editor and reviewers. The editor noted that the paper presents a “technological advance” that has enabled “important insights about the brain circuits through which the cerebellum could participate in social interactions.” Reviewer 1 thought this was a “timely and important study with solid evidence for correlative conclusions” and that the experiments were “technically challenging” and “well-performed”. Reviewer 2 stated that the finding of correlated activity between the regions is “interesting as non-motor functions of the cerebellum are relatively little explored.” They also thought “that the data are presented clearly, and the manuscript is well-written”. Reviewer 3 mentioned that “this approach can be useful for many neuroscientists”. We thank all the positive comments from the editors and all the reviewers.

**Reviewer #1 (Public Review)**
While the novelty of the device is strongly emphasized, I find that its value is somewhat diminished by the wire-free device developed by the same group as it should thus be possible to perform calcium imaging wire-free and electrophysiological recording via a single conventional cable (or also via wireless headstages).

While it would be potentially possible to use a wire-free Miniscope in parallel with a wired electrophysiology recording system, this would result in a larger footprint on the animal’s head, more than a gram in increased weight due to an added LiPo battery, a larger electrophysiology head-stage, and limited recording length due to a battery capacity of around 20 minutes. Our main goal for the development of the E-scope platform was to develop an expandable electrophysiology recording board that would work with all previously built UCLA Miniscopes while also streamlining the integration of power and data into the coaxial cable connection already familiar to hundreds of labs using Miniscopes. The vast majority of Miniscope experiments are done using wired systems and we aimed to support the expansion of those systems instead of requiring a more substantial switch to using wire-free Miniscopes.

The role of the identified network activations in social interactions is not touched upon.

We agree with the reviewer that we have not discovered a causal role for the co-modulated activity patterns we have observed. As these causal experiments will require the development of real-time techniques for blocking socially evoked changes in firing rate in cerebellum and ACC, we are currently planning experiments to address causality. These results will be described in a future publication.

**Reviewer #1 (Recommendations for the Authors):**
Please provide the number of recorded mice.

The number is now provided in the revised manuscript.

If the recorded areas (cerebellar cortex, DN, and ACC) are part of the same circuit regulating social interactions, it would be nice to get insights into the directionality of the circuit. The authors favor the possibility that during social behavior, cerebellar efferences indirectly influence ACC activities (as in Figure 4A), however, no evidence is presented to support this interpretation. ACC activities might also indirectly influence PC firing. It may be possible to get insights into this by comparing the timing of neuronal activity in the different areas with respect to social onset.

For this study, we mainly focused on the output of the cerebellar circuit to the cortex as previous work shows that dentate nucleus projects to the thalamus, which in turn projects to ACC and other cortical regions. (Badura et al.,eLife, 2018; Kelly et al., Nat. Neurosci., 2020) The temporal resolution of calcium imaging is limited (with the rise time of calcium events with genetically-encoded indicators taking hundreds of milliseconds) such that the resolution is insufficient to precisely assess the relative onset timing of the two regions. Our work certainly does not rule out cortical influences on PC firing.

**Reviewer #2 (Public Review)**
However, the causal relationship is far from established with the methods used, leaving it unclear if these two brain regions are similarly engaged by the behavior or if they form a pathway/loop.

As indicated in our response to Reviewer #1’s similar critique, the goal of the presented study is to demonstrate the feasibility and capabilities of this novel device. This new tool will allow us to conduct a comprehensive and rigorous study to assess the causal role of the interactions between the cerebellum and ACC in social behavior (as well as other behaviors). These experiments are being designed now.

**Reviewer #2 (Recommendations for the Authors):**
It is unclear what is entirely unique about the E-scope. It seems that its advance is simply a common cable that allows interfacing with both devices (lighter weight than two cables is stated in the Discussion). Is this really an advance? What are its limitations? E.g., how close can the recording sites be to one another? How can it be configured for any other extracellular recording approach (tetrodes, 64-channel arrays, or Neuropixels)?

In our experience, multiple lines of wires tethered to different head-mounted devices on an animal significantly impacts their behavior. Therefore, one of the major advantages of the UCLA Miniscope Platform is the use of a single, flexible coaxial cable to minimize the impact on tethering on behavior. The E-Scope platform builds on top of this work by incorporating electrophysiology recording capabilities into this single, flexible coaxial cable. Additionally, the electrophysiology recording hardware is backwards compatible with all previously built UCLA Miniscopes and can run through open-source and commercial commutators already used in Miniscope experiments.

The available bandwidth within the shared single coaxial cable can handle megapixel Miniscope imaging along with the maximum data output of a 32 channel Intan Ephys IC. The E-Scope platform presented here does run the Intan Ephys IC at 20KSps for all 32 channels instead of the maximum 30KSps due to microcontroller speed limitations, but this could be overcome by using a fast microcontroller or clock, or slightly reducing the total number of electrodes samples. Finally, the E-Scope was designed to support any electrode types supported by the Intan Ephys IC. This includes up to 32 channels of passive probes such as single electrodes, tetrodes, silicon probes, and flexible multi-channel arrays but does not include Neuropixels as Neuropixels use custom active electronics on the probe to multiplex, sample, and serialize electrophysiology data.

The authors only analyzed simple spikes in PCs for social-related activity. What about complex spikes? Is this correlated with ACC activity?

Complex spikes were detectable to the extent that we were able to define that the recorded cell was a PC, but because these cells were recorded in freely behaving mice, accurate complex spike detection was not reliable enough to be used for further correlational analyses.

The data is sampled in the two regions (cerebellum and ACC) at very different rates (imaging is much slower than electrophysiology; ephys data was binned). How does this affect the correlation plots?

We generated firing rate maps for the cerebellar neural activity using a binning size that matched the sampling frequency of calcium imaging (see Methods). As mentioned in our methods, to study the relationship between the electrophysiology and calcium imaging data we binned the spike trains using 33 ms bins to match the calcium imaging sampling rate (~30 Hz). This limits the temporal resolution to calculate fine-scale correlations, but the correlations that we report are on a behaviorally relevant temporal scale. The fine temporal resolution of the electrophysiology data however can still be used to further examine at a higher temporal resolution the relationship between cerebellar output and specific social behavior epochs.

For the correlation analysis, over what time frame was the activity relationship examined? How was this duration determined?

The main criteria for the time frame used to study the correlation analysis was the behavioral timescale of social interaction (see Author response image 1 for the number of social [red] and object [blue] interaction bouts [a], their duration [b] and coefficient of variation [CV] [c]). Overall, the activity relationship time frame was based on the average duration of the social interactions (~3 sec). Periods of 3.8 before and 5.8 sec after interaction onset were used to study. Accordingly, the cross-correlograms were constructed using a maximum lag length of 5 sec. In the article we reported correlation at lag 0.

**Author response image 1. sa4fig1:** 

The relationship between the cerebellum and ACC seems unconvincing. If two brain regions are similarly engaged by the behavior, wouldn't they have a high correlation? Is the activity in one region driving the other?

We reference studies showing an anatomical and functional indirect connection between the cerebellum and the ACC or prefrontal cortex (Badura et al., eLife, 2018). Also, as stated in the introduction, the ACC is a recognized brain area for social behavioral studies. In the results, we stated that correlations increase in groups of neurons that are similarly engaged during a specific epoch in the social interaction was an expected finding. What was not expected was that there would be no difference in the distribution's correlation when the social epochs were removed, suggesting that intrinsic connectivity does not drive a difference in correlations.

Although, since there is a cerebello-cortical loop, further study will be needed to understand which area initiates this type of activity during social behavior,

In the figures, the color-coded scale bars should be labeled as z-scores (confusing without them).In Figure 4, the color differences for Soc-ACC, Soc+ACC and SocNS ACC should be more striking as it is hard to tell them apart because they are all similar shades of blue-gray.

We thank the reviewer for their suggestions for improving the figures. We have incorporated these changes in Figures 2, 3 and along with their figure supplements. Graphs in Figure 4D-G have been edited to make the lines more visible to the reader.

**Reviewer #3 (Public Review)**
However, a mouse weighs between 20 and 40 g, so that an implant of 4.5 g is still quite considerable. It can be expected that this has an impact on the behavior and, possibly, the well-being of the animals. Whether this is the case or not, is not really addressed in this study.

The weight of the E-Scope (4.5 g) is near the maximum that is tolerated by animals in our experience. We therefore acclimated the mouse to the weight with dummy scopes of increasing weights over a 7-10 day period. During this period, we observed the animal to have normal exploratory behavior. Specifically, there is no change in the sociability of the animals (Figure 2A) and animals cover the large arena (48x 48 cm, Figure 2H).

Overall, the description of animal behavior is rather sparse. The methods state only that stranger age-matched mice were used, but do not state their gender. The nature of the social interactions was not described? Was their aggressive behavior, sexual approach and/or intercourse? Did the stranger mice attack/damage the E-Scope? Were the interactions comparable (using which parameters?) with and without E-Scope attached? It is not even described what the authors define as an "interaction bout" (Figure 2A). The number of interaction bouts is counted per 7 minutes, I presume? This is not specified explicitly.

As mentioned in the methods section of the original version of our manuscript, all the target mice were age-matched “male” mice. As per the reviewer’s suggestion, we now have added in the manuscript that before any of our social interaction behavioral experiments, aggressive or agitated mice were removed after assessing their behavior in the arena during habituation. For all trials, all mice were introduced for the first time.

We also mention in the methods section of our manuscript, that social behaviors were evaluated by proximity between the subject mouse and novel target mouse (2 cm from the body, head, or base of tail). From our recordings, we did not observe any aggressive, mounting, nor any other dominance behavior over the E-Scope subject mouse during the 7 minutes of social interaction assessment. Social interaction bouts in Figure 2A show the average number of social interaction bouts during the recording time. This has now been expanded upon in our revised manuscript.

It would be very insightful if the authors would describe which events they considered to be action potentials, and which not. Similarly, the raw traces of Figure 1E are declared to be single-unit recordings of Purkinje cells. Partially due to the small size of the traces (invisible in print and pixelated in the digital version), I have a hard time recognizing complex spikes and simple spikes in these traces. This is a bit worrisome, as the authors declare the typical duration of the pause in simple spike firing after a complex spike to be 20-100 ms. In my experience, such long pauses are rare in this region, and definitely not typical. In the right panel of Figure 1A, an example of a complex spike-induced pause is shown. This pause is around 15 ms, so not typical according to the text, and starts only around 4 ms after the complex spike, which should not be the case and suggests either a misalignment of the figure or the detection of complex spike spikelets as simple spikes, while the abnormally long pause suggests that the authors fail to detect a lot of simple spikes. The authors could provide more confidence in their data by including more raw data, making explicit how they analyzed the signals, and by reporting basic statistics of firing properties (like rate, cv or cv2, pause duration). In this respect, Figure 2 - figure supplement 3 shows quite a large percentage of cells to have either a very low or a very high firing rate.

We now provide a better example of simple spikes and complex spikes in Fig 1E and corrected our comment in the body of the manuscript. Previous version of the SS x CS cross-correlation histogram in Figure 1G as the reviewer mentions, was not the best example, because of the detected CS spikelets. However, the detection of CS spikelets has little impact on the interpretation of the results. We have replaced this figure with a better example of the SS x CS cross-correlation histogram.

The number of Purkinje cells recorded during social interactions is quite low: only 11 cells showed a modulation in their spiking activity (unclear whether in complex spikes, simple spikes or both. During object interaction, only 4 cells showed a significant modulation. Unclear is whether the latter 4 are a subset of the former 11, or whether "social cells" and "object cells" are different categories. Having so few cells, and with these having different types of modulation, the group of cells for each type of modulation is really small, going down to 2 cells/group. It is doubtful whether meaningful interpretation is possible here.

While the number of neurons is not as high as those reported for other regions, the number presented depicts the full range of responses to social behavior. It is extremely difficult to obtain stable neurons in freely behaving socially interacting animals and only a handful of neurons could be recorded in each animal. Among these recorded neurons only a subset responds to social interactions further reducing the numbers. The results however are consistent among cell types and the direction of modulation fits with the inhibitory connectivity between PCs and DN neurons. To our knowledge, we are the first group to publish neuronal activity of PC and DN neurons from freely behaving mice during social behavior.

Neural activity patterns observed during social interaction do not necessarily relate specifically to social interaction, but can also occur in a non-social context. The authors control this by comparing social interactions with object interactions, but I miss a direct comparison between the two conditions, both in terms of behavior (now only the number of interactions is counted, not their duration or intensity), and in terms of neural activity. There is some analysis done on the interaction between movement and cerebellar activity (Figure 2 - figure supplement 4), but it is unclear to what extent social interactions and movements are separated here. It would already help to indicate in the plots with trajectories (e.g., Fig. 2H) indicate the social interactions (e.g., social interaction-related movements in red, the rest of the trajectories in black).

We have updated the social interaction plots in Figure 2H in the revised version of the manuscript.

**Reviewer #3 (Recommendations for the Authors):**
Increase the number of cerebellar neurons that are recorded.

Due to the difficulty of the experiment and the low yield which we get for cerebellar recordings, substantially increasing the number of neurons will require many more experiments which are not feasible at this time.

Include more raw data and make the analysis procedure more insightful with illustrations of intermediate steps.

We have included a more thorough description of the analysis in the methods section of the revised manuscript.

Provide a better description of the behavior.

We have increased the level of detail regarding the mouse behavior in the Results and Methods sections. This includes a more detailed description of the parameters we used to analyze the social interaction.